Feeding behaviour in a ‘basal’ tortoise provides insights on the transitional feeding mode at the dawn of modern land turtle evolution

Natchev Nikolay 1 2 nikolay.natchev@univie.ac.at
Tzankov Nikolay 3
Werneburg Ingmar 4 5
Heiss Egon 6
1 Department of Integrative Zoology, Vienna University , Vienna , Austria
2 Faculty of Natural Science, Shumen University , Shumen , Bulgaria
3 Section Vertebrates, National Museum of Natural History, Bulgarian Academy of Sciences , Sofia , Bulgaria
4 Museum für Naturkunde, Leibniz-Institut für Evolutions- & Biodiversitätsforschung an der Humboldt-Universität zu Berlin , Berlin , Germany
5 Institut für Biologie, Humboldt-Universität zu Berlin , Berlin , Germany
6 Institute of Systematic Zoology and Evolutionary Biology, Friedrich-Schiller-University Jena , Jena , Germany
Anquetin Jérémy
Electronic publication date: 2015 Aug 11
Publication date: 2015
Volume: 3
Electronic Location ID: e1172
Received 2015 Mar 14; Accepted 2015 Jul 20
Copyright: © 2015 Natchev et al.
Copyright year: 2015
Copyright holder: Natchev et al.
License: This is an open access article distributed under the terms of the Creative Commons Attribution License, which permits unrestricted use, distribution, reproduction and adaptation in any medium and for any purpose provided that it is properly attributed. For attribution, the original author(s), title, publication source (PeerJ) and either DOI or URL of the article must be cited.
License URL: https://creativecommons.org/licenses/by/4.0/

Keywords: Food uptake, Turtle, Evolution, Tetrapoda, Feeding kinematics, Transition to land

Funding: FWF Austrian Science Fund P20094-B17 Swiss National Science Foundation P300P3 158526 The study was funded by FWF Austrian Science Fund, Project no. P20094-B17 on that EH and NN were employed and the Advanced Postdoc Mobility fund P300P3_158526 of the Swiss National Science Foundation granted to IW. The funders had no role in study design, data collection and analysis, decision to publish, or preparation of the manuscript.

==============================
Almost all extant testudinids are highly associated with terrestrial habitats and the few tortoises with high affinity to aquatic environments are found within the genus Manouria. Manouria belongs to a clade which forms a sister taxon to all remaining tortoises and is suitable as a model for studying evolutionary transitions within modern turtles. We analysed the feeding behaviour of Manouria emys and due to its phylogenetic position, we hypothesise that the species might have retained some ancestral features associated with an aquatic lifestyle. We tested whether M. emys is able to feed both in aquatic and terrestrial environments. In fact, M. emys repetitively tried to reach submerged food items in water, but always failed to grasp them—no suction feeding mechanism was applied. When feeding on land, M. emys showed another peculiar behaviour; it grasped food items by its jaws—a behaviour typical for aquatic or semiaquatic turtles—and not by the tongue as generally accepted as the typical feeding mode in all tortoises studied so far. In M. emys, the hyolingual complex remained retracted during all food uptake sequences, but the food transport was entirely lingual based. The kinematical profiles significantly differed from those described for other tortoises and from those proposed from the general models on the function of the feeding systems in lower tetrapods. We conclude that the feeding behaviour of M. emys might reflect a remnant of the primordial condition expected in the aquatic ancestor of the tortoises.

Introduction

Comprising more than 180 species, the cryptodiran taxon Testudinoidea represents the most diverse group of extant turtles (e.g., Fritz & Havaš, 2007; Thomson & Shaffer, 2010). Traditionally, it contains three major extant groups, including the emydids, the geoemydids and the testudinids (tortoises) (Fig. 1). All molecular phylogenetic studies (Iverson et al., 2007; Shaffer, 2009; Thomson & Shaffer, 2010; Barley et al., 2010) confirm a sister group relationship of the testudinids and the geoemydids (Fig. 1). The relationship of Platysternidae to other turtle groups remains unresolved (Parham, Feldman & Boore, 2006), but most molecular studies support a closer relationship to emydids (e.g., Thomson & Shaffer, 2010; Crawford et al., 2015). Palaeontological studies have shown that all testudinoids share aquatic ancestors, from which terrestrial species evolved (Danilov, 1999; Sukhanov, 2000; Joyce & Gauthier, 2004).

Figure 1 Phylogeny of turtle clades with a focus on Testudinoidea.

Interrelationship following Thomson & Shaffer (2010). Major evolutionary changes are listed; for details see text. Three modes of terrestrial food uptake are illustrated. (A) Jaw prehension; the tongue is not protruded and is only used for food transport (Geoemydidae, Manouria). (B) Jaw prehension; the elongated tongue is protruded during prehension but does not contact the food and is only used for food transport (Emydidae). (C) Prior to jaw prehension, the tongue contacts the food (advanced tortoises).

The majority of the Triassic stem turtles were terrestrial as indicated by the design and proportions of the limbs, which were adapted for terrestrial locomotion (for a comprehensive discussion see Joyce, 2015). With the emergence of modern turtles (Testudines) during the Jurassic period (e.g., Danilov & Parham, 2006; Sterli, 2010; Sterli & de la Fuente, 2011), a general transition of turtles into an aquatic environment occurred (Willis et al., 2013). The invasion of aquatic environments induced diversification into several subgroups (see Joyce, 2007; Thomson & Shaffer, 2010). Due to the different physical properties (drag, viscosity, etc.) of air and water, the new environment required morphological and functional adaptations of the locomotion and feeding system to enable efficient swimming behaviour and aquatic food uptake (i.e., suction feeding) (Schumacher, 1973; Lemell et al., 2002).

The sister group of all remaining Testudinoidea is the diverse and possibly paraphyletic extinct taxon †Lindholmemydidae (Fig. 1; Lourenço et al., 2012), which contains genera such as †Mongolemys and †Lindholmemys (Danilov, 1999; Joyce & Gauthier, 2004). Some poorly documented aquatic taxa such as †Haichemydidae and the †Sinochelyidae may perhaps also belong to †Lindholmemydidae. This group was recorded from aquatic sediments of the late Early Cretaceous and apparently had an amphibious lifestyle (Sukhanov, 2000). Among recent cryptodirans, semi-aquatic to semi-terrestrial lifestyle is typical for most emydids (plus platysternids) and geoemydids, as well as for some kinosternids (see Depeker et al., 2006, but also Nakajima, Hirayama & Endo, 2014). The remaining extant species are entirely terrestrial (tortoises), or predominantly aquatic (see Joyce & Gauthier, 2004; Rasmussen et al., 2011).

Among modern turtles, a transition from an aquatic to a semi-terrestrial or fully terrestrial habitat and the capacity to exploit terrestrial food sources has evolved independently within all three major testudinoid lineages (for overview see Summers et al., 1998; Natchev et al., 2009). At least eight emydid species are able to feed on land as well as under water (see Bels, Davenport & Delheusy, 1997; Bels et al., 2008; Summers et al., 1998; Stayton, 2011). During terrestrial feeding, such amphibious emydids use their jaws to grasp food items (jaw prehension). Similarly, all amphibious geoemydids studied to date also use jaw prehension in terrestrial food uptake (see Heiss, Plenk & Weisgram, 2008; Natchev et al., 2009). In contrast, all testudinids studied so far use the tongue to touch the food items, a behaviour referred to as “lingual prehension” (see Wochesländer, Hilgers & Weisgram, 1999; Bels et al., 2008). According to Bels et al. (2008), lingual prehension is obligatory for all tortoises.

The tortoises show a clear tendency towards herbivory and emancipation from water as living and feeding medium (see Pritchard, 1979; Ernst & Barbour, 1989; Bonin, Devaux & Dupre, 2006). In fact, testudinids seem to have lost their ancestral ability to feed under water and exclusively rely on terrestrial trophic ecologies. Some predominantly terrestrial geoemydids are able to complete the whole feeding process on land and under water (Natchev et al., 2010). Similarly, testudinids with tendencies towards an amphibious lifestyle might have retained the ancestral skill to feed underwater. Hence, information on bimodal feeding mechanisms in tortoises is of great importance to understand the evolution of terrestrial feeding mechanisms and subsequent evolution of the predominantly terrestrial lifestyle in tortoises.

The genus Manouria, being of the most ‘basal’ extant tortoises with a strong association to aquatic environments (Høybye-Mortensen, 2004; Stanford et al., 2015), constitutes a suitable model to study the feeding mechanisms in testudinids. Its partially aquatic feeding habit purported to be associated with the observed morphological extension of the palatines onto the triturating surface of the upper jaw (character 30 sensu Gerlach, 2001), a diagnostic feature common to geoemydid turtles. Another geoemydid-like feature is the unique existence of class II mental glands (Winokur & Legler, 1975).

The present study was conceived to provide a detailed analysis of the feeding behaviour in a species of the genus Manouria. Manouria emys is found in close association with water. Hence, we designed experiments to reveal whether this species is able to complete the entire feeding process under both aquatic and terrestrial conditions as some geoemydids do (see Natchev et al., 2009; Natchev et al., 2010).

Similar to all investigated testudinids, the Asian forest tortoise possesses a well developed tongue. The hyoid complex is predominantly cartilaginous (Heiss et al., 2011). On the base of the specific morphology of the feeding apparatus (elastic basis of the oropharynx and voluminous lingual structures) we suggest a poor suction feeding performance in case Manouria attempts to feed under water.

Wochesländer, Hilgers & Weisgram (1999), Wochesländer, Gumpenberger & Weisgram (2000) and Bels et al. (2008) stated that the feeding kinematics in all testudinids involve two common features: an obligatory lingual prehension and the split of the gape cycle in four main phases: slow open phase I (SOI); slow open phase II (SOII); fast open phase (FO); fast close phase (FC). In our experiments we test whether these kinematical elements are present in the feeding behaviour of M. emys. On the basis of our findings, we fine-tune the kinematical feeding models proposed for tortoises. The gained new data requires a re-evaluation of the concept on the function of the tongue in food uptake in tortoises. Having in mind the phylogenetical position of M. emys and the specifics of its feeding behaviour, we propose a hypothesis on the evolution of the terrestrial feeding among testudinoids in particular and turtles in general. We discuss also the interrelationship between the diet and the feeding media in the course of turtle evolution.

Materials and Methods

Ecological background

Both extant species of Manouria, the Asian forest tortoise M. emys and the impressed tortoise M. impressa, have a restricted distribution in Southeast Asia. M. emys has a narrow distribution in Bangladesh, India (Assam, Meghalaya, Mizoram, Nagaland), Myanmar, Thailand, Malaysia (East and West), and Indonesia (Kalimantan, Sumatra). The nominate subspecies, M. emys emys—the subject of this study, inhabits the southern part of the species range (Fritz & Havaš, 2007; Stanford et al., 2015).

M. emys inhabits tropical evergreen monsoon forests and exhibits high tolerance for soil moisture. It is commonly found reposing in wet areas, buried in mud or under the leaf litter where it may spend long periods of time. It is active even during rainy weather. Direct sun exposure and basking are not required. Furthermore, this species has a mostly crepuscular and nocturnal lifestyle (Ernst, Altenburg & Barbour, 2000; Vetter & Daubner, 2000; Stanford et al., 2015).

According to the available literature, the diet of M. emys includes plants, fungi, invertebrates, and frogs (Nutphand, 1979; Das, 1995; Lambert & Howes, 1994; Høybye-Mortensen, 2004). It has been reported to feed on plants in shallow mountain streams (Nutphand, 1979).

Experimental setting

Animal husbandry and experiments were in strict accordance with the Austrian Protection of Animals Act. The animals used in the present study were obtained commercially and kept at 12 h dark/light cycles in a large terrarium (150 × 100 cm ground area) with a permanently filtered water basin and spacious terrestrial area. The turtles were fed different fruits, vegetables, commercially obtained tortoise pellets, dead mice, as well as pieces of cattle heart and liver, offered on the terrestrial part of the terrarium. Carapace lengths in the three subadult experimental animals ranged between 109 and 135 mm with body masses between 234 and 236 g. For filming terrestrial feeding, the specimens were put in a dry glass cuvette (24 × 60 × 30 cm). When the food was offered on the floor of the cuvette, the tortoises often twisted their necks and rotated their heads in attempt to grasp the food item. The side movements made the filming of the animals in strict lateral view very difficult and the landmarks were not clearly visible during the sequence. By the use of forceps for food display we completely eliminated these problems and were able to shoot perfect lateral plans of the feeding turtles. The food in the feeding experiments was offered at a position which was similar to the position on which we offered the food in the terrarium where the tortoises were housed. The animals did not extend vastly their necks to reach the food items (see Appendix S1). The position of the offered food was completely “natural”. The tortoises needed to stretch their necks forwards rather than downwards, which did not impacted other kinematic patterns of the feeding cycles.

As food items we used small pieces of cattle heart measuring approximately 5 × 5 × 5 mm. The turtles were filmed from lateral aspect (with a reference grid 1 × 1 cm in the background) via the digital high-speed camera system Photron Fastcam-X 1024 PCI (Photron limited, Tokyo, Japan) at 500 fps with a highly light-sensitive objective AF Zoom—Nikkor 24–85 mm (f/2, 8-4D IF). Two “Dedocool Coolh” tungsten light heads with 2 × 250 W (ELC), supplied by a “Dedocool COOLT3” transformer control unit (Dedo Weigert Film GmbH, München, Germany) were used for illumination. We filmed and analysed the food uptake and the food transport cycles in eight feeding sequences for each specimen.

The setting for filming aquatic feeding of submerged food comprised the experimental aquarium filled with water to a level of 3 cm and presentation of food items in front of the turtle’s snout. In order to reduce the light intensity and for optimisation of the digitising process, the frame rate was reduced to 250 fps. As the tortoises were unable to grasp the food item in a total of 36 trials, the kinematics of the feeding apparatus had been analysed (see below) in nine selected representative feeding trials.

For both terrestrial and “aquatic feeding” sequences, horizontal (X-axis) and vertical (Y-axis) coordinates of relevant landmarks (see Fig. 2) were digitised frame by frame using “SIMI-MatchiX” (SIMI Reality Motion Systems, Unterschleißheim, Germany). Based on the displacement of the markers, we were able to calculate the gape amplitude (distance between the tip of the upper and lower beak), head movement (distance between the anterior tip of the carapace and the point “P” on Fig. 2), tongue movements (distance between the most ventral point on tympanum and the tip of the tongue when visible), and hyoid movements (distance between the point “P” on Fig. 2 and the basis of the posterior ceratobranchial). To compare the kinematic feeding pattern of M. emys to those of other studied turtles and to understand the coordination between the elements of the feeding apparatus, these data were used for calculation of the following kinematical variables: duration of Slow open phase (SO); duration of Slow open phases I and II (SOI and SOII) when present; duration of fast open (FO); duration of maximum gape phase (MG); duration of fast close (FC); time to peak gape (TPG); total cycle duration (TCD); duration of hyoid protraction (HyDD); duration of hyoid retraction (HyVD); duration of the total hyoid cycle (THC); hyoid retraction velocity (HyRV); duration of head protraction (HP); duration of head retraction (HR); duration of tongue protraction (TP); tongue retraction velocity; delay of the start of hyoid retraction relative to the tongue retraction start; delay of reaching peak gape relative to start of the hyoid retraction; delay of reaching peak gape relative to tongue retraction start (see Table 1).

Figure 2 Selected frame from a high-speed video sequence (500 frs) of food transport in Manouria emys, showing the landmarks used for kinematic analyses.

C, rostral tip of sagital line of the carapace; Hy, hyoid at the basis of ceratbranchial I; LJ, tip of the lower jaw; P, posterior most point of crista supraoccipitale; TT, tip of the tongue; Tv, ventral most point of the tympanum at the position of the jaw joint; UJ, tip of the upper jaw; grid 10 × 10 mm. Abbreviations in Appendix S1.

Table 1 Variables describing the feeding process in Manouria emys, present as means ± SD; n, sample size.

Abbreviations in Appendix S1.

	Food uptake (FU)	Transport (T)	I vs. T	
Variable	Individual 1	Individual 2	Individual 3	p1	Individual 1	Individual 2	Individual 3	p2	p3	
	(n = 8)	(n = 8)	(n = 8)		(n = 33)	(n = 20)	(n = 21)			
SOI duration (s)	0.168 ± 0.060	0.618 ± 0.231	0.562	n.c.	0.146 ± 0.016	0.126 ± 0.014	0.115 ± 0.015	0.378	0.068	
	n = 2	n = 3	n = 1		n = 18	n = 14	n = 11			
SOII duration (s)	0.738 ± 0.508	0.453 ± 0.294	1.024	n.c.	0.147 ± 0.014	0.187 ± 0.027	0.190 ± 0.021	0.187	0.072	
	n = 2	n = 3	n = 1		n = 18	n = 13	n = 11			
FO duration (s)	0.450 ± 0.060	0.379 ± 0.150	0.694	n.c.	0.122 ± 0.009	0.126 ± 0.012	0.102 ± 0.006	0.111	0.011*	
	n = 2	n = 3	n = 1		n = 25	n = 15	n = 19			
MG duration (s)	0.079 ± 0.017	0.095 ± 0.031	0.166 ± 0.044	0.271	0.025 ± 0.003	0.033 ± 0.004	0.042 ± 0.019	0.318	0.001*	
	n = 4	n = 4	n = 6		n = 6	n = 10	n = 4			
FC duration (s)	0.157 ± 0.079	0.105 ± 0.036	0.158 ± 0.030	0.024*	0.089 ± 0.020	0.186 ± 0.040	0.119 ± 0.088	0.155	0.010*	
	n = 8	n = 8	n = 8		n = 33	n = 20	n = 21			
TPG (s)	0.943 ± 0.144	0.989 ± 0.177	1.784 ± 0.137	0.002*	0.408 ± 0.021	0.439 ± 0.038	0.403 ± 0.028	0.187	<0.001*	
	n = 8	n = 8	n = 8		n = 33	n = 20	n = 21			
TCD duration (s)	1.139 ± 0.148	1.128 ± 0.169	2.073 ± 0.144	0.001*	0.499 ± 0.020	0.655 ± 0.098	0.510 ± 0.030	0.311	<0.001*	
	n = 8	n = 8	n = 8		n = 33	n = 20	n = 21			
HDD duration (s)					0.281 ± 0.025	0.216 ± 0.035	0.169 ± 0.019	0.005*		
					n = 30	n = 14	n = 21			
HVD duration (s)					0.176 ± 0.011	0.167 ± 0.014	0.149 ± 0.009	0.162		
					n = 31	n = 17	n = 21			
THC duration (s)					0.456 ± 0.028	0.384 ± 0.041	0.317 ± 0.022	0.002*		
					n = 30	n = 14	n = 21			
HRV velocity (cm/s)					0.718 ± 0.059	0.938 ± 0.107	0.551 ± 0.071	0.016*		
					n = 31	n = 17	n = 21			
HP duration (s)	1.345 ± 0.159	1.204 ± 0.246	2.494 ± 0.177	0.001*	0.220 ± 0.049	0.864 ± 0.132	0.464 ± 0.089	0.001*	<0.001*	
	n = 8	n = 8	n = 8		n = 32	n = 14	n = 10			
HR duration (s)	0.296 ± 0.041	0.487 ± 0.086	0.704 ± 0.174	0.052	0.236 ± 0.027	0.211 ± 0.025	0.316 ± 0.065	0.333	0.002*	
	n = 8	n = 7	n = 8		n = 14	n = 13	n = 10			
TP duration (s)					0.165 ± 0.008	0.160 ± 0.017	0.133 ± 0.025	0.483		
					n = 30	n = 19	n = 13			
TR velocity (cm/s)					7.459 ± 0.550	5.798 ± 0.547	6.562 ± 0.595	0.121		
					n = 31	n = 20	n = 13			
Delay of HVD start relative to TR start (s)					−0.2011 ± 0.026	−0.039 ± 0.231	−0.082 ± 0.025	0.005*		
					n = 30	n = 15	n = 14			
Delay of TPG relative to HVD start (s)					−0.007 ± 0.007	−0.016 ± 0.008	−0.032 ± 0.005	0.014*		
					n = 31	n = 17	n = 21			
Delay of TPG relative to TR start (s)					−0.062 ± 0.007	−0.045 ± 0.013	−0.055 ± 0.008	0.521		
					n = 31	n = 20	n = 14			
Notes.

* Significant differences (α = 0.05) among individuals in the ingestion phase (P1), in the transport phase (P2), and between both mode (P3); n.c., p value not calculated.

Statistics

We tested for any differences among the frequency of occurrences of defined patterns both in food uptake (FU) and food transport (T), i.e., sequences with: missing split of the jaw opening in SO and FO; without detectable split of discrete SOI and SOII slow gape phase; lacking MG phase. In order to provide the comparisons, Chi-square test with Yates’ correction was performed. Then we tested for possible existence of differentiation in kinematical variables in both feeding stages (FU and T). All variables were tested with the Shapiro–Wilk test for normal distribution. When the p-value was less than the chosen alpha level (p < 0.05), the null hypothesis was rejected and data were excluded from further analyses. In addition, all variables included in Table 1 were tested with Levene’s and Brown–Forsythe tests and then processed with Welch’s ANOVA for heteroscedastic data. Tukey’s honest significant difference test (HSD) was performed for post-hoc analyses when applicable.

Furthermore, in order to express the degree of individual differentiation among the studied specimens, a Canonical discriminant analysis (CDA) was performed. Standard descriptive statistics including mean, range, standard deviation ((SD) and confidential interval at 95% CI) were presented.

Results

When feeding on land, the Asian forest tortoises always grasped food by the jaws. After food uptake, one to four transport cycles followed prior to oesophageal packing (see Schwenk, 2000). The tip of the tongue was barely visible during food uptake (see Figs. 3B and 3C) indicating that the tongue was not protracted. By contrast, during transport cycles, the cyclic movements of the tongue were well visible as it was rhythmically pro- and retracted to transport the food item towards the oesophagus (Fig. 5).

Figure 3 Selected frames and graphics (based on a high-speed video with 500 frs) represent the movements of jaws, hyoid and t head during terrestrial food uptake in Manouria emys when feeding on pieces of beef heart.

(A) slow open phase end (lacking discrete SOI and SOII); (B) fast open end; (C) fast close start; (D) fast close end; arrows indicate the position of the food item; arrowheads represent the position of the tip of the tongue; grid 10 × 10 mm. Abbreviations in Appendix S1.

When trying to feed under water (Fig. 4 and at http://figshare.com/s/5d9e23c8f4ec11e49cb306ec4b8d1f61), M. emys submerged its head under the water level and protruded the gaping jaws toward the food item. The gape cycle was newer split in slow and fast jaw open phases. The tongue tip was not visible from the lateral aspect and the hyolingual complex did not protract prior reaching peak gape. No retraction of the hyoid complex was detected prior jaw closure. The gape cycle duration exceeded one and a half seconds and was 1.94 ± 0.36 s (mean ± SD). Despite the unsuccessful attempts, the turtles repeatedly tried to catch the submerged food. In several events, we were able to detect that the food item was carried away by the bow wave induced by jaw closing.

Figure 4 Selected frames and graphics (based on high-speed video with 250 frs) showing the movements of jaws, hyoid, and head during attempts of aquatic food uptake in Manouria emys.

(A) start of the gape cycle; (B) end of jaw opening; (C) maximum gape end; (D) fast closure end; note the lack of movement of the hyoid complex during the whole cycle; grid 10 × 10 mm. Abbreviations in Appendix S1.

Figure 5 Selected graphics (based on a high-speed video with 500 fr/s) showing the movement patterns of jaws, hyoid, tongue and head during terrestrial food transport in M. emys; note the delay in hyoid ventral displacement relative to the start the retraction of the tongue tip, as well as the delay of both the tongue retraction and hyoid retraction relative to the start of the FO phase.

Abbreviations in Appendix S1.

Figure 6 Graphical representation of three selected variables in food uptake (FU) and food transport (T) phases.

Bars are denoted by their mean values and whiskers present the 95% CI; (A) sequences with discrete SO and FO phases; (B) sequences with no detectable split of discrete slow gape phase (SOI and SOII were inseparable); (C) sequences with lacking MG phase. Abbreviations in Appendix S1.

The variables of the kinematical profiles are summarised in Table 1. In the statistic tests, we found highly significant differences in sequences with and without both SOI and SOII when food uptake and transport stages were compared χ1,N=982=25.05,p<0.001. Similarly significant differences were observed when comparing food uptake and transport cycles in respect to sequences with and without slow jaw open phases as well as with and without maintaining jaw maximum gape—MG phase χ1,N=982=6.10,p=0.02;χ1,N=982=6.52,p=0.01.

Nine of the variables which describe the food uptake process were detected to show significant differences between individuals (Table 1). In transport cycles, six out of 18 variables differed significantly amongst individuals (see Table 1). Seven out of nine variables differed significantly when testing for differences between grasping and transport cycles: fast jaw open duration (FO; FWelch(1,43) = 15.17, p = 0.011); maximum gape (MG; FWelch(1,26) = 15.89, p = 0.001); fast closing (FC; FWelch(1,26) = 7.86, p = 0.010); time to peak gape (TPG; FWelch(1,72) = 46.78, p < 0.001); total gape cycle duration (TCD; FWelch(1,72) = 52.50, p < 0.001); head protraction duration (HP; FWelch(1,67) = 52.23, p < 0.001); and head retraction duration (HR; FWelch(1,47) = 12.57, p = 0.002).

When comparing three further parameters among the transport cycles in all three specimens (delay of HyVD start relative to TR start; delays of TPG relative to HyVD; delay of TPG relative to TR starts), statistically significant differences were found among all compared pairs (FWelch(2,105) = 41.58, p < 0.001).

The performed canonical discriminant analysis (CDA) revealed the existence of substantial degree of individualism among the studied specimens (Fig. 7). However, only the first axis eigenvalue exceeded the level of acceptance, i.e., 1. First axis explained 70% of the total variance. Among the 18 studied variables only two (THC and HPR) showed higher correlation scores than 0.75. On the base of the CDA and the detected degree of individualism mentioned above, we can conclude that the patterns displayed by the studied specimens can be regarded as similar but not as uniform.

Figure 7 Canonical centroid plots of three Manouria emys specimens (T1–T3), centroid scores for each individual and measurement repetition in food transport phase.

Discussion

The Asian forest tortoise repetitively tried to feed on dispersed food items under water, which was an unexpected and hitherto unknown behaviour among tortoises. However, M. emys always failed to consume the submerged food. On land, M. emys grasped food with the jaws, just like all known aquatic or semiterrestrial turtles do, but not with the tongue as formerly predicted for all tortoises. On the basis of our results we discuss several important evolutionary, behavioural, and functional aspects.

Evolution of food uptake among turtles

In general, most aquatic turtles combine a fast acceleration of the head towards the food or prey item and a suction feeding mechanism is induced by fast oropharyngeal volume expansion. In some extant turtles, a strong suction flow can be generated and prey is directly sucked into the oropharynx without contact with the jaws (e.g., Chelus fimbriatus (Lemell et al., 2002), Apalone spinifera (Anderson, 2009), Pelodiscus sinensis (N Natchev & I Werneburg, 2013, unpublished data)). However, most extant turtles cannot generate such strong suction flows and only compensate (“gulp”) the bow wave that otherwise would push small to moderately sized food items away from the fast approaching head. These species finally fix and grasp prey with the jaws (see Lauder & Prendergast, 1992; Lemell, Beisser & Weisgram, 2000; Aerts, Van Damme & Herrel, 2001; Natchev et al., 2009; Natchev et al., 2011). We consider the latter plesiomorphic behaviour for extant turtles.

Among extant turtles, the ability to complete the whole feeding process (including food uptake, food manipulation and transport, esophageal packing, and swallowing) on land has been tested and documented for only six species so far. All of them were members of Testudinoidea (Fig. 1; see also Summers et al., 1998; Bels et al., 2008; Natchev et al., 2009). The terrestrial mode of food uptake differs dramatically among and within the three testudinoid subgroups (see Bels, Davenport & Delheusy, 1997; Bels et al., 2008; Summers et al., 1998; Wochesländer, Hilgers & Weisgram, 1999; Natchev et al., 2009, present study). Correspondingly, it appears as if terrestrial feeding re-evolved several times independently amongst turtles. Unfortunately, only limited experimental data are available on feeding mechanisms in emydids and geoemydids. Further functional and palaeontological investigations may help to sort out the issues on the evolution of the feeding behaviour and the morphology of the feeding apparatus in testudinoids.

Very limited information is available on feeding mechanisms employed by amphibious non-testudinoid turtles that occasionally exploit terrestrial food sources. Weisgram (1985a) and Weisgram (1985b) documented a kinosternid (Claudius angustatus) that caught prey on land and dragged it into water for transport and swallowing. Natchev et al. (2008) documented another kinosternid (Sternotherus odoratus) catching food on land, but failing to transport it through the oropharynx. Among extant turtles, successful food transport on land seems to be restricted to testudinoids. The development of enlarged and muscular tongues within this group (Von Bayern, 1884; Werneburg, 2011) represents adaptation to improved terrestrial food manipulation.

Based on experimental data, Natchev et al. (2009) described and summarised three categories of terrestrial food uptake modes among Testudinoidae: (A) Jaw prehension with retracted hyolingual complex, as observed in the geoemydid genus Cuora (Natchev et al., 2009); (B) Jaw prehension with slightly protracted hyolingual complex, as observed in emydids (Bels, Davenport & Delheusy, 1997; Stayton, 2011); (C) Lingual prehension—the tongue touches the food item prior to food uptake, as documented in all tortoises studied so far (Wochesländer, Hilgers & Weisgram, 1999; Bels et al., 2008). The food uptake mode of M. emys, however, differs substantially from that of all remaining tortoises (category C). In fact, the hyolingual complex in M. emys remained fully retracted during the food prehension on land, and the first contact with the food item was by the jaws. Accordingly, the feeding mechanism of M. emys should be assigned to category A, along with that of semi-aquatic geoemydids.

We now aim to construct a theoretical scenario on the evolution of terrestrial feeding mechanisms in turtles. Given the aquatic origin of all living turtles, the functional transition from aquatic to terrestrial feeding mechanisms could hypothetically have involved four stages, beginning with an exclusively aquatic feeding ancestor. In different lineages and stages, turtles may have left their aquatic environments for various reasons e.g., for exploiting new food niches. The species that retained predominantly aquatic life styles may grasp food by the jaws on land, but have to drag it into the water for further intraoral (hydrodynamic based) transport. In recent turtles, such behaviour was documented in the kinosternids C. angustatus (Weisgram, 1985a; Weisgram, 1985b) and S. odoratus (Natchev et al., 2011), as well as in the emydid Trachemys scripta (Weisgram, 1985b; Weisgram, Dittrich & Splechtna, 1989) and other emydids (see Stayton, 2011). Turtles of the second hypothetical evolutionary stage grasped food by the jaws, while the tongue was used for intraoral food transport on land. Such species would still have retained their underwater feeding ability by using hydrodynamic mechanisms. When grasping food on land, the tongue remained retracted or was protracted without touching the food item. Among extant turtles, such a feeding mode is found in the geoemydid genus Cuora (Heiss, Plenk & Weisgram, 2008; Natchev et al., 2009; Natchev et al., 2010) and in some emydids (Bels, Davenport & Delheusy, 1997; Summers et al., 1998; Stayton, 2011). In the next theoretical evolutionary step (stage three), behavioural and morphological adaptations for terrestrial feeding were further advanced, increasing the efficiency of terrestrial food transport at the expense of the ability to use effective hydrodynamic mechanisms in water. Such species still grasped food items with their jaws on land (as typical for aquatic or semiaquatic turtles), but were no longer able to take up dispersed food if submerged, which features prominently in the present case of M. emys. Finally, in a fourth stage, turtles became fully terrestrial and their tongue was obligatorily involved in food uptake as documented in the tortoises Testudo (Eurotestudo) hermanni boettgeri (Weisgram, 1985b; Wochesländer, Hilgers & Weisgram, 1999), Kinixis belliana, Geochelone elephantopus and G. radiata (Bels et al., 2008).

Our investigations demonstrate that the ‘basal’ tortoise M. emys does not contact food with the tongue prior to jaw prehension on land. This shows that tongue to food contact is characteristic of advanced tortoises only. We consider the terrestrial feeding behaviour of M. emys as plesiomorphic and potentially inherited from its semiaquatic ancestors. On that basis, M. emys can be considered a transitional turtle in regard to secondary terrestriality.

We propose that hyolingual protrusion evolved in the lineage forming to advanced tortoises (Fig. 1). Manouria emys has a large tongue with massive intrinsic and extrinsic musculature (see Heiss et al., 2011). The advanced and complex lingual musculo-skeletal architecture allows the turtle to protrude the tongue outside the margins of the rhamphothecae (see Fig. 2). However, M. emys does not use lingual food prehension as typical for all other tortoises studied so far. In fact, it seems that the Manouria (and perhaps Gopherus (N Natchev, pers. obs., 2015)) “lineage” has retained the jaw prehension mechanism inherited from earlier aquatic ancestors. It seems that the tortoises, in general, evolved fleshy tongues which improve the food transport performance. The advanced tortoises only refined the behaviour of food uptake on land via lingual food contact prior to jaw closure (see Wochesländer, Hilgers & Weisgram, 1999; Bels et al., 2008).

Function of the protruded tongue in the testudinid’s food uptake

What would be the potential advantage of the obligatory lingual protrusion, found in the more derived tortoises? One possible explanation is that the tongue is used as a prehensile organ for food ingestion analogous to that found in other tetrapod groups (for overview see Schwenk, 2000; Schwenk & Wagner, 2001). However, for tortoises such interpretation might be put into question. By examining all published data available (Wochesländer, Hilgers & Weisgram, 1999; Bels et al., 2008), we were not able to find any convincing evidence that tortoises collect food with their tongues—they just touch it. In all published feeding sequences, the contact between the food and the tongue is clearly demonstrated—yet, in all cases, after the initial contact of the tongue with the food, the head moves forward and the food item is not dragged into the mouth by tongue retraction, but is grasped by the jaws during the fast jaw closing (FC gape phase). Initial food ingestion in tortoises might not be considered “lingual prehension” (see Schwenk, 2000; Bels et al., 2008) in the strict sense, but should be regarded as “jaw prehension following lingual contact”. This prompts the question: why is an obligatory contact of the tongue to the food present in extant tortoises (except in Manouria and also conceivably in Gopherus) during food uptake? In other words: why do tortoises apply a more complex and presumably more energetically expensive food uptake mechanism by including movements of the hyolingual complex in addition to the movements of the neck and jaws alone?

We propose that the lingual contact provides tactile information on the position of the food item and helps the advanced tortoises to compensate the “information gap” which occurs when the food is approached to a distance where it is out of sight. The eyes of tortoises are positioned laterally on the head (Pritchard, 1979) and the turtles are not able to permanently observe the position of the food item when the neck is protracted and the gape is positioned around it. The prolonged maximum gape (MG) phase found in most ingestion cycles of M. emys (see Table 1, Figs. 3 and 6) might be the result of lack of lingual contact with the food surface. In all published sequences and kinematical profiles on food uptake in tortoises, there is a clear tendency toward a split of the gape cycle into slow open (SO) and fast open (FO) gape phases (see Wochesländer, Hilgers & Weisgram, 1999; Bels et al., 2008). The lack of tongue protrusion might explain the lack of slow open (SO) and fast open (FO) split in the gape cycle of food uptake in the geoemydid Cuora (see Natchev et al., 2009). In most food uptakes analysed in M. emys, SO phases are not present and the gape increases gradually (see Table 1 and Fig. 6). Similar as in Cuora, the lack of SO phases might be explained by the lack of tongue protrusion in food uptake (see Natchev et al., 2009).

Intraoral food transport on land

The execution of the transport cycles require coordination of the activities of contractile elements such as the jaw opening and closing muscles, head protracting and retracting muscles, intrinsic and extrinsic lingual muscles, as well as muscles that protract and retract the hyolingual complex as a whole unit (Jones et al., 2012; Werneburg, 2011; Werneburg, 2013). In contrast to this complicated choreography, the mode of food prehension in M. emys suggests less complex neuromotoric coordination between neck and jaw movements. Yet, the execution of the transport cycles is often more than twice shorter in duration (see Table 1). A possible explanation for the longer duration of food uptake cycles relative to transport cycles might be that during transport, the coordination centres of the muscle activities execution are permanently supplied with information concerning the position of the food item within the oropharynx and the proper movements can be executed precisely in a shorter time.

In his work on the feeding mechanisms in domestic animals, Bels (2006) established that a pre-programmed Generalise Cyclic Model (GCM)—very similar to those proposed from Bramble & Wake (1985)—is universally valid among the different groups of tetrapods. The feeding kinematics of the Asian forest tortoise differs in some aspects from those proposed by the GCM. The kinematics of the feeding system in M. emys seems to be pre-programmed, but under permanent feedback control. The values of the gape and hyoid/hyolingual cycle patterns in the three specimens studied here show high degrees of variation, both concerning food uptake and food transport (see Table 1 and Figs. 5–7). The slow open phases (SO) are not obligatory. The gape cycle often includes a phase of retaining maximum gape (see Table 1, Figs. 3 and 5). In turtles, the maximum gape phase (MG) was described for the gape cycle in Cuora sp. (Natchev et al., 2009; Natchev et al., 2010) and was confirmed for kinosternidae (Natchev et al., 2011). The presence of a prolonged maximum gape phase (MG) can be easily overlooked when the frame rate of the film sequence is not high enough (i.e., step between successive frames over 10 ms). Thus, it may be present in other turtles, but was not taken into account by the calculations of the kinematical profile (see Stayton, 2011; Nishizawa et al., 2014).

The GCM presupposes that the start of hyoid retraction coincides with the start of fast open phase (FO). However, our calculations (see Table 1 and Fig. 5) demonstrate that in M. emys the hyoid retraction in the food transport cycle starts shortly prior reaching peak gape. The same pattern was detected by the investigation of aquatic, semi-aquatic, but also predominantly terrestrial cryptodirans (Natchev et al., 2008; Natchev et al., 2009; Natchev et al., 2010; Natchev et al., 2011).

Relations between the habitat preferences and the diet in turtles

In the evolution of the testudinids there is a clear shift not only in the habitat preferences (from aquatic to terrestrial), but corresponding shifts are also seen in dietary preferences. In that taxon it manifests in a tendency toward herbivory. Most of the recent tortoises rely on diets mainly composed of plant material (for overview see Pritchard, 1979; Ernst, Altenburg & Barbour, 2000; Bonin, Devaux & Dupre, 2006). The partly carnivorous lifestyle of Manouria sp. (Bonin, Devaux & Dupre, 2006) may be a relic of the carnivorous diet of the ancestor of the tortoises and supports the transitional status of the genus.

Apparently the feeding media (air vs. water) and the dietary shift had a large influence on the overall feeding behaviour of the testudinids (see Bels et al., 2008). The suction mechanism was lost and replaced by a jaw food prehension system (this study) or by “lingual prehension” (Wochesländer, Hilgers & Weisgram, 1999; Bels et al., 2008). Probably, the switch to herbivory determined the reorganisations in the morphology of the jaw muscle system and the proportions of the skull in tortoises (see Werneburg, 2011; Werneburg, 2012; Werneburg, 2013). By feeding predominantly on plants (immobile items), the tortoises may prolong the duration of the food uptake cycles and have more time to adjust their prehension kinematics to every single feeding situation.

We propose that the habitat preferences and the diet change in turtles are firmly correlated (e.g., Bels et al., 2008; Werneburg, 2014). The overall rigid design of the turtle ‘body plan’ hinder the animals to actively hunt for agile prey in terrestrial environments (King, 1996). We hypothesise that this statement is also valid for the terrestrial stem turtles and that these animals were predominantly herbivorous (sensu King, 1996). In aquatic turtles, in contrast, the buoyancy of the water overrides the shell-caused restrictions of mobility and also suction feeding can be applied for carnivorous feeding. Those advantages might have been the key factor for the turtles to become aquatic (and carnivorous) in the Jurassic. There may be several reasons for the secondary terrestriality in tortoises and presumably one of the main reasons was the inter- and intraspecific concurrence for food resources.

Conclusions

We propose that the ancestral food uptake mode in tortoises was jaw based when feeding on land. During the shift from aquatic to terrestrial lifestyle, including a shift from aquatic to terrestrial feeding biology, the Manouria (and most likely also Gopherus) “lineage” had retained pure jaw prehension in food uptake. The hyolingual complex in that lineage exhibits the typical morphological features of tortoises that feed exclusively on land (see Bramble, 1973; Winokur, 1988; Heiss et al., 2011), such as an enlarged fleshy tongue with abundant papillae, a complex tongue musculature, a relatively small and mainly cartilaginous hyoid and hypoglossum. The evolutionary shift in the morphology of the hyolingual complex was apparently primed by the optimisation of the food transport behaviour and not for food uptake. We suggest that the involvement of the tongue during food uptake found in the derived extant tortoises serves as a tactile sensory tool for the localisation of the food item prior to jaw prehension. Thus, the tongue is not used as the main food collecting organ in modern tortoises and the food uptake mode represents a derived jaw prehension system.

Supplemental Information

Appendix S1 Appendix

Click here for additional data file.

We would like to thank Josef Weisgram, Andreas Wanninger, Patrick Lemell, Christian Beisser and Thomas Schwaha (Department for Integrative Zoology, University of Vienna) for providing material and suggestions for the executions of our experiments. Alexander Westerström contributed sorely to the revision of our manuscript. Stefan Kummer, Katherina Singer, Monika Lintner and Marion Hüffel are acknowledged for the careful housing of the animals. The reviewers provided helpful comments and suggestions to improve our paper.

Additional Information and Declarations

Competing Interests

Author Contributions

Animal Ethics

Data Availability

The authors declare there are no competing interests.

Nikolay Natchev conceived and designed the experiments, performed the experiments, analyzed the data, contributed reagents/materials/analysis tools, wrote the paper, prepared figures and/or tables, reviewed drafts of the paper.

Nikolay Tzankov and Ingmar Werneburg analyzed the data, wrote the paper, prepared figures and/or tables, reviewed drafts of the paper.

Egon Heiss conceived and designed the experiments, performed the experiments, analyzed the data, contributed reagents/materials/analysis tools, wrote the paper, reviewed drafts of the paper.

The following information was supplied relating to ethical approvals (i.e., approving body and any reference numbers):

The animals were commercially obtained. As no invasive techniques were applied, no approval documents were required by the Austrian Protection of Animals Act at the time of the experiments.

The following information was supplied regarding the deposition of related data:

Figshare: figshare.com/s/5d9e23c8f4ec11e49cb306ec4b8d1f61 & figshare.com/s/1920fc6af4ed11e4972106ec4bbcf141.

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
