# Peer review of "Feeding behaviour in a ‘basal’ tortoise provides insights on the transitional feeding mode at the dawn of modern land turtle evolution"

_PeerJ, doi:10.7717/peerj.1172_

## Round 0.1 · original submission · Major Revisions

The two reviewers agree that your study is interesting and constitutes an important contribution to the field. However, each of them points out several issues in your experimental design (statistics part, presentation of food by forceps) and interpretations (change of diet in advanced tortoises, generalized tetrapod feeding cycle, hyoid movements) that need to be addressed before your manuscript could be considered for publication.

Another important issue that needs to be addressed is the organisation of your manuscript. I share the opinion of both reviewers that your manuscript, notably the discussion part, needs to be revised and perhaps shortened in order to improve its readability and comprehension. Please, heed their remarks in this matter. When all the above is done, I also recommend that you carefully proofread the text and figures (notably figure 1) for typos, grammatical errors, and awkward word choices.

Reviewer 1 ·

Basic reporting

First, there are both grammatical and organization issues with the writing in this manuscript. The grammatical issues are not too bad – the authors (perhaps not native English speakers) should simply run the paper through another round of editing and revisions to ensure that everyone is appropriately constructed. Typically issues have to do with definite articles and word choices. But there are also frequent problems with spacing and other mechanical issues that need to be addressed.

More importantly, the authors need to revise the paper for better organization. I found myself quite confused at many parts of this paper. In some places, paragraphs have too many topics and need to be broken up (lines 32-55). In other places, the length is appropriate, but the topics are not intuitively organized (lines 56-68, 377-395). It seems to me as though the authors need to spend more time in the Introduction discussing the four-phase model of feeding kinematics, as well as its variations – this, rather than the aquatic feeding abilities of M. emys, seems to be where the authors focus most of their analysis (given that the tortoises never successfully fed underwater). The “Ecological background” section of the Materials and Methods should be moved to the introduction. The Results section is a good length, but I think that the Discussion could be considerably shortened.

I didn't make editorial annotations on the manuscript, but I have included a list of minor editorial comments as a .pdf.

Experimental design

My one major conceptual criticism with the paper has to do with the perceived causes of the changes in kinematics observed between the various turtles, including tortoises. The authors assume that all of the changes that occur in the feeding kinematics of the lineages leading to terrestrial turtles occurred due to the transition from water to land. This is reasonable, but these lineages also made a major transition from feeding primarily on animals to feeding on plants. I’d like the authors to address the issue of how they separate a sequence of changes that were induced by a change in feeding medium from a sequence of changes that might be induced from a major change in diet. This is particularly acute for this paper, as Manouria emys is both more aquatic than most tortoises, and more carnivorous.

Much more attention also needs to be given to the “Statistics” portion of the Materials and Methods. I don’t understand the first sentence, even after reading the entire manuscript. And the rest of the section is also confusing. There a number of ways to fix this – the authors could state the null hypotheses being tested, or the questions being addressed, for each test, for example. Either way, I need a lot more information and more clarity in this section – what is currently present is not sufficient. I am also concerned about some of the analyses – isn’t it inevitable that kinematic differences would be found between sequences with a SO I/II phase, and those without, given that the phases are defined kinematically? Perhaps I’m misunderstanding here, but in any case it needs to be explained more fully.

In am not sure why the authors perform a CDA, as they don’t really reference it or use it in the discussion of their results. And the analysis itself is not sufficiently described in the Materials and Methods. This could probably be deleted, but if the authors choose to keep it, they should revise their discussion of the axes – it’s not possible for the first 2 axes to explain 100% of the variation among individuals (70% for the first seems fine), so did the authors mean something else?

Finally, the authors need to make some major revisions to their figures. Figure 1, 2, and 4 are okay, though I would like the authors to explain why they showed a different hypothesis for the position of Platysternon, and what the dotted lines are supposed to represent. But arrows and arrowheads are not found in all parts of figure 3, and I’m afraid that I found Figure 5 impossible to interpret. The abbreviations used are very confusing (“H” should not stand for both “hyoid” and “head”) and not all listed in the caption are found in the figure (I couldn’t find “GCM” and “THC”). In Figure 6 – A, B, and C should be replaced with more meaningful descriptions (“Split between SO and FO” for “A”) and the axis needs to be labeled.

Validity of the findings

The authors make certain assertions that are simply incorrect. Tortoises are not the only pure terrestrial living turtle group (line 31) – many box and wood turtles (Terrapene, Cuora, Rhinoclemmys, Glyptemys) are just as terrestrial as many tortoises. And I wouldn’t say that tortoises show a “broad variety (line 41) in their feeding ecology – they’re all herbivores, sometimes with omnivorous tendencies, and there really aren’t any non-herbivores specialists or any with special feeding kinematics. In other cases, the authors make assertions that really ought to be phrased as hypotheses – the authors don’t know that the specimens were “trying to position the gaping mouth around the food item” (line 144). Similarly, the ideas that “Tortoises in general evolved fleshy tongues…and advanced tortoises only refined the behavior of food uptake…” are hypotheses, not established facts. I wouldn’t call the radiation of aquatic turtles “immense” (it’s maybe 250 species, yes?) or say that they have had “great evolutionary success” (lines 201 and 202) and I’m a turtle biologist. The authors should be careful to indicate when they are proposing hypotheses or interpreting data, and when they are stating observed facts.

Additional comments

The study is well-grounded. M. emys is an important tortoise to study, given its phylogenetic placement and unconventional (for tortoises) environmental and dietary preferences. And the results are certainly interesting – all criticisms (see below) aside, the authors’ findings that M. emys individual do not show feeding behaviors and kinematic patterns that were assumed to be ubiquitous for all tortoises will certainly be of interest to a wide variety of researchers. I was also very interested in the four-stage model that the authors propose for the evolution of terrestrial feeding in turtles. So I think that this work constitutes an important contribution to the field. However, there are a number of issues that the authors need to address to maximize the utility of this manuscript.

The paper needs to be revised, with particular attention to mechanics and organization. The authors should describe why they think that the feeding behaviors that they observe related to a change from an aquatic to a terrestrial lifestyle, and not to a change from carnivory to herbivory. The statistics need to be more fully described, and the figures need to be revised.

Annotated reviews are not available for download in order to protect the identity of reviewers who chose to remain anonymous.

·

Basic reporting

The paper is generally well written, but could benefit from a review by a native speaker. There are enough akward phrases, typos and errors to distract the reader from the content of the manuscript. Some eamples are listed below:

line 41: emancipation

line 47 & 51: ambiguous

line 81: depending of sun basking (basking implicitly implies an animal being in the sun)

there are many others throughout.

I believe that the manuscript could be more focused in places (especially in the discussion) and woudl suggest the authors to have a good look at the discussion and to take out everything that does not contribute to the principal question addressed. This will make the paper easier to read.

Experimental design

The basic experimental design is good. I do have, however, one major issue. The prey were presented by forceps to the animals in a terrestrial setting. This is very un realiistic and may notcorrespond at all to natural feeding. The unusual gape cycles observed during capture may be due to the fact that prey were presented above the animal and not on the bottom of the cage where animals would typically eat prey from. This may also bias teh aquatic-terrestrial comparisons. The authors need to show that the presentation of food via forceps above the animal does not affect the kinematics before the data can be interpreted the way they are.

Validity of the findings

The findings are generally valid but see the caveat reported higher. However in the interpretation of the results on lines 309 the authors refer to the Bramble and Wake model and go on to discuss that no tongue-food contact takes place during capture. Although this is true for the species they studied, The Bramble and Wake model did not discuss prey capture but only intraoral transport. As such the authors should compare transport cycles to the B&W model and not their capture cycles. The transport cycles do resemble a typical 'generalized tetrapod feeding cycle'. This needs to be changed.

Finally on line 401 the authors discuss hyoid movements based on external video - they really cannot do this even if I agree that there is some correspondence between the movement of the mouth floor and the hyoid. They really need videofluoroscopy to do this. This part of the discussion shoudl be removed.

Additional comments

Overall an interesting manuscript that brings novel data.

---

## Round 0.2 · Minor Revisions

I agree with Reviewer 2 that you have made significant improvements to your manuscript. However, the first Reviewer considers that several changes that were suggested during the first round of review still need to be addressed. I would like to give you the opportunity to reply specifically to these suggestions, either making appropriate changes in the manuscript or offering a response to the reviewer’s concerns.

Concerning issues with spelling and grammar, I have proofread your manuscript and suggested some changes. There are also several references that are missing from the reference list. See attached annotated manuscript.

Reviewer 1 ·

Basic reporting

I am afraid that there are still significant problems with the writing in this manuscript. In some cases there are errors in spelling (“linage” rather than “lineage” in lines 322, 327, and 428, “Suplements” rather than “Supplements”, line 135) or in the standard use of idioms in English (entities are only prohibited “from” activities, not “to” them, line 416; “belike” is not a standard phrase, line 348). But more often there are problems with the grammar and organization of the manuscript. The following are some of the majors problems that remain with grammar: The authors have a tendency to write run-on sentences: “A and B”, where “A” and “B” are complete sentences that should be written separately (lines 45-48, 113-114, 135-137, 294-295). The order of information was odd for the description of the results of the statistical analyses – rather than “When testing/comparing…x, y, and z were found”, the authors should simply write “X, y, and z were found when testing…” or even “X, y, and z were found” (207-215).

The manuscript still seems poorly organized. I would prefer to see hypotheses listed at the end of the Introduction – it is not sufficient to state that hypotheses are formulated (98), or to provide a sentence from which hypotheses can be inferred (90-92). There is still material in the Discussion that I think should go in the intro – in particular, more background on the evolution of terrestriality in testudinoids (256-265), the standard model of amniote feeding (311-316), the GCM (379-381), and so on. The Introduction still gives the reader very little idea of what to expect in this manuscript. Some information in the Introduction is superfluous or insufficiently described in order for the reader to understand its importance (84-88).

I still fail to see the importance of the CDA. Very little information has been added to the manuscript. Currently, the CDA is presented as an illustration of the variation in feeding kinematics of the specimens used in this study. However, this variation was not a subject of study, and is only relevant for statistical tests which are already covered. I still maintain that this analysis is unnecessary and should be removed from the manuscript.

Additionally, some assertions are simply incorrect. The terrestrial testudinoids did not all evolve directly from the common aquatic ancestor of testudinoids, of course, or at least not any more than any of the aquatic testudinoid turtles. Instead, the terrestrial lineages each evolved from a separate aquatic lineage. “Geoemydid” is not synonymous with “batagurid” (86). Manouria is not the only basal tortoise (97). Tortoises do not permanently twist their necks while feeding (129).

Experimental design

The description of the statistical tests has been improved, though the authors should clarify what are the two feeding stages that are being compared (is it the food uptake and food transport stages?). I still think a statement of hypotheses being tested using the statistics would be useful. And I still wonder whether it is at all noteworthy that the authors found kinematic differences between parts of the feeding cycle that are defined, at least partially, by kinematic differences.

Validity of the findings

Two major issues remain regarding interpretation of the results. The first regards potential causes of the differences in feeding kinematics between aquatic and terrestrial species. I suggested that the herbivorous diets of tortoises and other terrestrial testudinoids, rather than their terrestrial habitats, might be the cause of the changes in kinematics. The authors now address this (414-422), and I agree with them that the rigid body plan of turtles probably requires that primarily terrestrial lineages also evolve to be primarily herbivorous. But that still does not help to determine whether tortoises feed the way that they do because they are terrestrial, or because they are herbivores. For this, I might suggest comparisons with aquatic herbivores (among testudinoids, many Pseudemys and some Graptemys would be appropriate, or even Dermatemys outside of Testudinoidea) – has anyone looked at the feeding kinematics of these species.

Second, there are still a few hypotheses being presented as observations – the authors do not know that the animals were “trying to position the gaping mouth around the food item” (line 194). This speculation should be identified as such.

I will also note that the authors statement that stage 3 of their proposed evolutionary sequence represents a “point of no return” for terrestrial feeding (306). Clearly it is possible for fully terrestrial taxa to evolve the ability to feed in aquatic environments, if the original turtles were terrestrial (53).

·

Basic reporting

No comments

Experimental design

No comments

Validity of the findings

No comments

Additional comments

I have now read the revised version and I'm happy with the revisions. The authors have done a good job in streamlining the paper and it has become much clearer I believe.

---

## Round 0.3 · accepted · Accept

You have revised your manuscript according to suggestions and I can now formally accept it for publication. Congratulations!

I have noted the two following minor issues, which can be addressed at proof stage: 1/ reference for Joyce (2015) is not complete; 2/ Weisgram et al. (1989) cited line 288 is still not in the reference list.